# Evidence for sample selection effect and Hawthorne effect in behavioural HIV prevention trial among young women in a rural South African community

Molly Rosenberg,[1] Audrey Pettifor,[2,3,4] Rhian Twine,[4] James P Hughes,[5] F Xavier Gomez-Olive,[4,6,7] Ryan G Wagner,[4,8] Afolabi Sulaimon,[4] Stephen Tollman,[4,7,8] Amanda Selin,[3] Catherine MacPhail,[4,9,10] Kathleen Kahn[4,7,8]

## ABSTRACT

**Objectives** We examined the potential influence of both sample selection effects and Hawthorne effects in the behavioural HIV Prevention Trial Network 068 study, designed to examine whether cash transfers conditional on school attendance reduce HIV acquisition in young South African women. We explored whether school enrolment among study participants differed from the underlying population, and whether differences existed at baseline (sample selection effect) or arose during study participation (Hawthorne effect).

**Methods** We constructed a cohort of 3889 young women aged 11–20 years using data from the Agincourt Health and socio-Demographic Surveillance System. We compared school enrolment in 2011 (trial start) and 2015 (trial end) between those who did (n=1720) and did not (n=2169) enrol in the trial. To isolate the Hawthorne effect, we restricted the cohort to those enrolled in school in 2011.

**Results** In 2011, trial participants were already more likely to be enrolled in school (99%) compared with non-participants (93%). However, this association was attenuated with covariate adjustment (adjusted risk difference (aRD) (95% CI): 2.9 (– 0.7 to 6.5)). Restricting to those enrolled in school in 2011, trial participants were also more likely to be enrolled in school in 2015 (aRD (95% CI): 4.9 (1.5 to 8.3)). The strength of associations increased with age.

**Conclusions** Trial participants across both study arms were more likely to be enrolled in school than non-participants. Our findings suggest that both sample selection and Hawthorne effects may have diminished the differences in school enrolment between study arms, a plausible explanation for the null trial findings. The Hawthorne-specific findings generate hypotheses for how to structure school retention interventions to prevent HIV.

## Strengths and limitations of this study

► To our knowledge, this study is the first to empirically examine whether Hawthorne effects may have influenced study results in an HIV prevention trial.
► We analysed longitudinal data on a key study outcome (school enrolment) for the underlying population from which study participants were drawn. Complete data are not typically available for source populations in research studies.
► Our Hawthorne-specific findings suggest that aspects of the HIV Prevention Trial Network (HPTN) 068 protocol could potentially be adapted for school retention interventions to prevent HIV.
► It is important to note that data on HIV incidence, the primary endpoint of HPTN 068, were not available for the underlying target population.
► The differences we attribute to the Hawthorne effect were estimated in an observational data set with adjustment for key sociodemographic characteristics. The potential for uncontrolled confounding requires that our results be interpreted cautiously.

participant characteristics differ from those in the target population, even with randomisation of exposure (referred to here as 'sample selection effect').[3] Further threats to validity can occur if study participation itself induces behaviour change (Hawthorne effect, research participation effect or trial effect, referred to here, collectively, as 'Hawthorne effect').[4 5] Analysing study data to examine how results may have differed in the target population to which we would like to make inference is critical to making valid conclusions and policy recommendations.

Although epidemiological training and research have long included at least cursory examinations of external validity,[6] with more recent methodological advancements around transportability of effect estimates from study

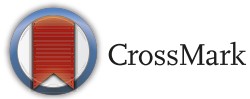

For numbered affiliations see end of article.

**Correspondence to**
Dr Molly Rosenberg;
rosenmol@indiana.edu

## INTRODUCTION

The evidence base for public health interventions largely comes from rigorous epidemiological studies.[1 2] However, results from epidemiological studies may not be generalisable (or 'externally valid') when study

populations to target populations,[7–9] empirical evaluation of Hawthorne effects is rare. The limited evidence for Hawthorne effects comes largely from clinical randomised controlled trials, most often assessed in cancer and nutrition studies,[10 11] with supporting evidence from HIV treatment research.[12] Trials designed to affect behaviour change may be particularly susceptible to Hawthorne effects as the behaviours in question may be influenced by trial participation.[13 14] This is particularly true in HIV prevention research, where Hawthorne effects could pose validity threats if unexpectedly low HIV incidence occurs due to trial-induced risk behaviour changes.[15] To our knowledge, no prior HIV prevention trial has empirically examined whether Hawthorne effects influenced study results.

In this study, we examine the potential influence of both sample selection effects and Hawthorne effects in the behavioural HIV Prevention Trial Network (HPTN) 068 study,[16 17] designed to examine whether cash transfers conditional on school attendance reduce HIV acquisition in young South African women. Contrary to the study hypothesis, no difference in HIV acquisition was observed between study arms, with high levels of school enrolment and low HIV incidence in both arms. These findings were surprising given the high background rates of school dropout in the study area[18–20] and the large body of evidence showing the positive impact of cash transfers on schooling outcomes,[21] and limited the ability to explore schooling as a mechanism to reduce HIV risk.

Here, we contextualise HPTN 068 findings, using data on school enrolment in the underlying target population routinely collected by the Agincourt Health and socio-Demographic Surveillance System (HDSS) in which HPTN 068 was nested. We examine whether school enrolment trajectories of trial participants differed from non-participants, and whether differences could be attributed to existing differences in school enrolment at baseline (sample selection effect) or differences that arose during study participation (Hawthorne effect).

## METHODS
### Study setting and population
The Agincourt HDSS is located in the rural Bushbuckridge municipality of the Mpumalanga province, South Africa,[22] and has routinely collected annual vital event data on all people living in the study area since 1992.

Other sociodemographic data are collected at regular but less frequent intervals. For example, educational attainment is queried every 3 years, employment data are collected every 4 years, and a household asset index is measured every other year. Community, household and individual consents have been obtained for all Agincourt HDSS census research since its inception, with informed verbal consent obtained at each census round. The Agincourt HDSS currently surveys the full cohort of over 115 000 people living across 31 villages, in an area of economic disadvantage with historically low access to public services. However, government schools in the study site are free and often provide feeding programmes. HIV contributes a large burden to the community, with 19% HIV prevalence overall in those aged 15 years and older.[23]

HPTN 068 was a phase III individually randomised trial designed to examine whether cash transfers conditional on school attendance influence the risk of HIV acquisition in young women.[16 17] Young women and their caregivers were randomly assigned to receive a monthly cash transfer conditional on ≥80% school attendance or no cash transfer. The size of the monthly cash transfer was 300 rands (R; about US$30 in 2012), and was divided into R200 provided to the caregiver and R100 provided to the young woman. Key selection criteria for participation in the study were current enrolment in grades 8–11; age 13–20 years; not married or pregnant at baseline; and having a caregiver with the documents necessary to open a bank account. Age-eligible young women were identified from Agincourt HDSS records to be contacted for further eligibility screening (n=10 134).[16] Between March 2011 and December 2012, a total of 2533 young women were enrolled. Participants were seen annually for a maximum of 3 years from enrolment or until high school graduation. Thus, participants who enrolled in the trial in 2011 in grade 11 could exit the study as early as 2012 after graduating high school. Participants who enrolled in the trial in 2012 in grade 8 or 9 could exit the study as late as March 2015 (figure 1).

### Cohort construction
We constructed our analytical cohort to identify all young women living in the study area at the time of trial start (2011) regardless of trial participation status. Further restrictions were applied to build a cohort of young women on comparable age/grade trajectories and to match key HPTN 068 selection criteria. First, we restricted

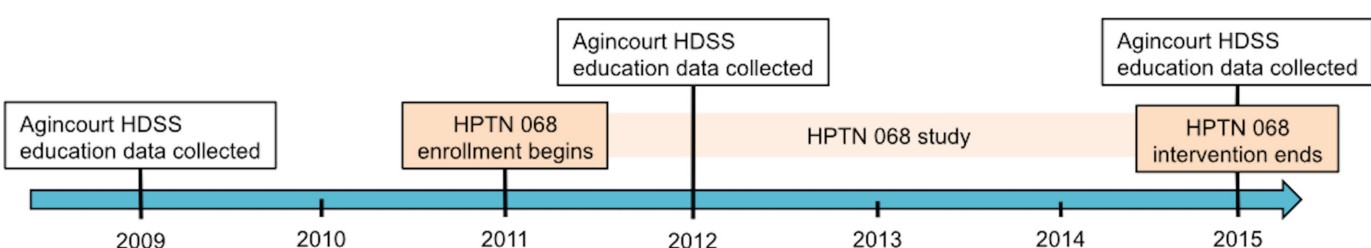

**Figure 1**  Timeline of Agincourt HDSS education data collection and HPTN 068 trial duration. HDSS, Health and socio-Demographic Surveillance System; HPTN, HIV Prevention Trial Network.

the cohort to include young women between the ages of 13 and 20 years in 2011 or 2012. Based on education data collected by the Agincourt HDSS in 2009 (education data were collected in 2009 but not again until 2012), we also restricted the cohort to those who were enrolled in grades that projected to grades 8–11 in 2011 or 2012, assuming a one-grade increase each year.

## Key measures

Our primary exposure of interest was *HPTN 068 trial participation* (both trial arms combined). We analysed both arms together because trial results indicated essentially no differences in school attendance and enrolment data between the arms. School attendance was high (≥80%) for 95% of the intervention arm and 96% of the control arm participants. Permanent school dropout occurred at a rate of 3 per 100 person years in both arms.[17]

Our primary outcome of interest was *school enrolment*, which we calculated at 2011 and 2015 based on Agincourt HDSS education data collected in 2009, 2012 and 2015. We used the 2011 school enrolment outcome to assess whether enrolment patterns were already different at the beginning of the trial, indicating a potential sample selection effect. We used the 2015 school enrolment outcome to assess whether enrolment patterns were different at the end of the trial, when both sample selection and Hawthorne effects could be present. We considered young women as enrolled in school if they indicated current school enrolment or if they reported a grade 12 attainment, the final year of secondary schooling.

With the 2009, 2012 and 2015 education modules, fieldworkers updated the highest education level each young woman achieved and recorded whether or not she was currently enrolled in school. The 2015 time-point aligned with a census education module, so enrolment status decisions were made based on data reported at that time. The 2011 time-point did not align with a census education module, so we inferred enrolment status at this time-point based on changes in status between 2009 and 2012. If young women reported enrolment in both 2009 and 2012, we inferred they were also enrolled in 2011. If young women reported enrolment in 2009 but not 2012, we used changes in educational attainment to infer whether school dropout had occurred before or after 2011. For example, if a young woman reported grade 7 attainment in 2009 and grade 10 attainment in 2012, we assumed that the 3 years of additional education were accumulated in 2009, 2010 and 2011. This young woman would be coded as enrolled in school in 2011. If a young woman reported grade 7 attainment in 2009 and grade 9 attainment in 2012, we assumed that the 2 years of additional education were accumulated in 2009 and 2010. This young woman would be coded as not enrolled in school in 2011. Observations with illogical education data patterns between 2009 and 2012 (eg, educational attainment decreases over time) were coded as missing for school enrolment in 2011.

We also explored the potential for confounding and effect measure modification by key covariates. We examined age on 1 January 2011, categorised as ages of compulsory school enrolment (ages 11–15 years), older than age for compulsory school enrolment but correct age for grade (ages 16–17 years), and older than age for compulsory school enrolment while also older than expected for grade (ages 18–20 years). We also examined indicators of household socioeconomic status (SES), measured with a composite index of household assets; household size; gender of household head; secondary school educational attainment of the household head; country of origin (South African or Mozambican descent); and pre-2011 childbearing.

## Analysis

We used binomial regression models with an identity link to estimate the association between trial participation and school enrolment in 2011 and 2015. The 2011 enrolment outcome was used to isolate the potential for a sample selection effect (ie, Were trial participants more likely to be in school than non-participants at the beginning of the trial?). We used the 2015 enrolment outcome in a restricted cohort of young women who were enrolled in school in 2011 to isolate the potential for a Hawthorne effect at the end of the trial (ie, Were trial participants more likely to remain in school after 4 years than non-participants, conditional on being in school at the beginning of the trial?).

We conducted unadjusted analyses and analyses adjusted for age, SES, gender and education of household head, household size, country of origin, and pre-2011 childbearing. School enrolment decisions were likely highly influenced by age both because our cohort straddled the age limit for compulsory schooling in South Africa and because school dropout generally increases with age. Thus, we conducted age-stratified analyses to see whether the associations between trial participation and school enrolment differed by age category.

Although trial results indicated that school attendance and enrolment outcomes were not significantly different between the arms of the trial, the intervention was designed to incentivise school attendance.[17] For this reason, we conducted a sensitivity analysis restricting the trial participants to those who were randomly assigned to the control group only. We compared results from this restricted population with the results from the analysis of the full trial population.

## RESULTS

Overall 3889 young women from the Agincourt HDSS were included in our cohort (table 1). The median age was 15 years (IQR: 14–16). Young women tended to live in large households (mean size: 8.4), and household heads were often female (42%) and often lacked high school education (86%). The majority of young women were of South African descent (60%) and very few (7%) had

Table 1 Sociodemographic characteristics of a cohort of 3889 young women between the ages of 11 and 20 years in rural Agincourt, South Africa, by participation in the HPTN 068 trial

| | Total (n=3889) | | Trial participant* (n=1720) | | Non-participant (n=2169) | | |
|---|---|---|---|---|---|---|---|
| | n | % | n | % | n | % | P value† |
| **Age in 2011** | | | | | | | <0.0001 |
| 11–12 | 188 | 4.8 | 26 | 1.5 | 162 | 7.5 | |
| 13–14 | 1270 | 32.7 | 637 | 37.0 | 633 | 29.2 | |
| 15–16 | 1591 | 40.9 | 793 | 46.1 | 798 | 36.8 | |
| 17–18 | 667 | 17.2 | 239 | 13.9 | 428 | 19.7 | |
| 19–20 | 173 | 4.5 | 25 | 1.5 | 148 | 6.8 | |
| **SES‡** | | | | | | | 0.003 |
| At or above median | 1778 | 50 | 853 | 52.7 | 925 | 47.8 | |
| Below median | 1778 | 50 | 766 | 47.3 | 1012 | 52.3 | |
| Missing | 333 | | 101 | | 232 | | |
| **Household size‡** | | | | | | | 0.5 |
| Mean | 8.4 | | 8.4 | | 8.5 | | |
| SD | 4.2 | | 4.0 | | 4.4 | | |
| Missing | 46 | | 14 | | 32 | | |
| **Gender of household head‡** | | | | | | | 0.3 |
| Female | 1590 | 41.7 | 691 | 40.8 | 899 | 42.4 | |
| Male | 2224 | 58.3 | 1004 | 59.2 | 1220 | 57.6 | |
| Missing | 75 | | 25 | | 50 | | |
| **Household head educational attainment‡** | | | | | | | 0.02 |
| <Grade 12 | 3062 | 86.3 | 1398 | 87.8 | 1664 | 85.1 | |
| Grade 12 or higher | 486 | 13.7 | 194 | 12.2 | 292 | 14.9 | |
| Missing | 341 | | 128 | | 213 | | |
| **Country of origin‡** | | | | | | | 0.2 |
| South Africa | 2315 | 59.6 | 1046 | 60.9 | 1269 | 58.6 | |
| Mozambique | 1570 | 40.4 | 673 | 39.2 | 897 | 41.4 | |
| Missing | 4 | | 1 | | 3 | | |
| **Childbearing before 2011** | | | | | | | <0.0001 |
| Yes | 274 | 7.1 | 67 | 3.9 | 207 | 9.5 | |
| No | 3615 | 93.0 | 1653 | 96.1 | 1962 | 90.5 | |
| **Intervention arm** | | | | | | | |
| Control | | | 820 | 50.4 | | | |
| Intervention | | | 806 | 49.6 | | | |
| **2011 school enrolment** | | | | | | | <0.0001 |
| Yes | 3508 | 95.7 | 1637 | 99.2 | 1871 | 92.9 | |
| No | 158 | 4.3 | 14 | 0.9 | 144 | 7.2 | |
| Missing | 223 | | 69 | | 154 | | |
| **2015 school enrolment** | | | | | | | <0.0001 |
| Yes | 2465 | 74.2 | 1234 | 19 | 1231 | 68.5 | |
| No | 856 | 25.8 | 290 | 81 | 566 | 31.5 | |
| Missing | 568 | | 196 | | 372 | | |

*Due to restrictions in the cohort construction to maintain comparable groups with respect to age and education status in 2009, not all of the 2533 HPTN 068 participants are represented.
†P values for categorical variables are from $X^2$ tests and for continuous variables from t-tests.
‡Measured in 2009.
HPTN, HIV Prevention Trial Network; SES, socioeconomic status.

begun childbearing prior to 2011. Just under half of the young women (44%) went on to participate in HPTN 068, and they tended to be less likely to be on the youngest (ages 11–12) or oldest (ages 19–20) end of the age spectrum, although the median age of both participants and non-participants was 15. Trial participants were also less likely to have begun childbearing, slightly more likely to live in households headed by high school graduates and slightly more likely to live in household with above-median SES.

In 2011, at the time of HPTN 068 trial start, nearly everyone in the cohort was enrolled in school (96%), likely due to the age and 2009 enrolment requirements placed on the cohort (table 2). However, young women who became trial participants were already more likely to be enrolled in school (99%) compared with non-participants (93%) (risk difference (95% CI): 6.3 (5.1 to 7.5)), indicating a sample selection effect likely occurred as a consequence of the school enrolment eligibility criterion. Although the overall association attenuated after covariate adjustment (adjusted risk difference (aRD) (95% CI): 2.9 (−0.7 to 6.5)), strong associations were observed in the age group of 18–20 years old (aRD (95% CI): 19.5 (5.6 to 33.3)), compared with the younger two age groups ($aRD_{11-15}$ (95% CI): 0.4 (−5.2 to 5.9); $aRD_{16-17}$ (95% CI): 6.3 (0.5 to 12.2)).

At the end of the trial in 2015, the difference in school enrolment between trial participants (81%) and non-participants (69%) grew, with an adjusted overall risk difference of 6.8 (95% CI 3.4 to 10.2). To investigate whether any differences in school enrolment could be attributed to a Hawthorne effect, we restricted the cohort to those enrolled in school in 2011 and examined differences in 2015. Under this restriction, young women enrolled in the trial were still more likely to remain in school in 2015 (82%), compared with those who did not (74%), with an aRD of 4.9 (95% CI 1.5 to 8.3)). Again, the association was weakest among those 11–15 years old (aRD (95% CI): 1.8 (−1.3 to 5.0)) and strongest among those 18–20 years old (aRD (95% CI): 22.8 (7.3 to 38.2)).

Results were largely unchanged when we restricted the trial participant population to those assigned to the control group only (table 3). Although CIs widened due to reduced sample size with some newly spanning the null, the magnitudes of the risk difference point estimates were largely unchanged from those in the primary analysis.

## DISCUSSION

HPTN 068 found that cash transfers conditional on school enrolment did not influence HIV acquisition among young women in a rural South African setting. Due to unexpectedly high levels of school enrolment in both arms, the ability to explore schooling as a mechanism through which cash transfers could influence HIV acquisition was limited. Here, we found evidence to suggest that both Hawthorne effects and sample selection

effects could threaten the external validity of these findings. Overall, trial participants were more likely to remain in school until graduation than non-participants. Differences in school enrolment status were already apparent at the beginning of the study, suggesting that the trial selection criteria likely pulled in young women with better school enrolment behaviours than those who were not enrolled as trial participants (sample selection effect). Differences in school enrolment grew larger as the trial progressed, and importantly remained strong even after restricting to those enrolled in school in 2011 when the trial started, suggesting the changes in enrolment status occurred during the trial itself (Hawthorne effect). Both sample selection and Hawthorne effects may have diminished the differences in school enrolment between study arms and is one plausible explanation for the overall null study effect. The HPTN 068 trial was designed to activate the HIV prevention effects of education by incentivising school attendance and retention in the intervention arm. With high levels of school attendance and retention across both arms of the trial, the ability to detect a trial effect was likely weakened.

Our findings that trial participation influenced school enrolment behaviour could plausibly be explained by several characteristics of the HPTN 068 study design and protocol. First, all participants were aware of the objective of the study: to retain young women in school to prevent HIV. This information could result in school enrolment behaviour change to align with perceived expectations of study staff or because young women were motivated to prevent HIV. Second, compared with non-participants, trial participants were exposed to different networks likely to be supportive of school enrolment. Adult fieldworkers showed interest in the schooling of participants with yearly in-person data collection and monthly in-school data collection. Data were collected in 'camps' wherein trial participants were transported to study offices annually for a half-day of surveys and blood tests, and entertaining activities during wait periods (eg, fingernail painting, photograph taking, magazine reading). This protocol could have fostered a cohesive group environment among trial participants resulting in positive peer pressure to maintain school enrolment. There is a growing body of evidence that interventions providing a safe space with adult mentorship and peer support can have positive outcomes for young women in sub-Saharan Africa,[24–26] a pathway that may have been activated with trial participation. Finally, trial participation provided access to certain health and social services that may have otherwise been inaccessible, including annual HIV and herpes simplex virus type 2 (HSV-2) testing and counselling, linkage to care for those who tested positive, and linkage to social work services for young women who reported experiences of sexual abuse. These services may have enabled young women who would have otherwise struggled with serious physical and mental health outcomes to remain in school.

Associations between trial participation and school enrolment were strongest in older age groups. The small

**Table 2** The relationship between HPTN 068 trial participation and school enrolment in 2011 and 2015, stratified by age, in the full cohort and the cohort restricted to those enrolled in school in 2011

| | Enrolment | n | Per cent enrolled | RD (95% CI) | aRD* (95% CI) |
|---|---|---|---|---|---|
| **2011: full cohort** | | | | | |
| All ages | | | | | |
| Trial participant | 1637 | 1651 | 99.2 (98.7, 99.6) | 6.3 (5.1 to 7.5) | 2.9 (−0.7 to 6.5) |
| Non-trial participant | 1871 | 2015 | 92.9 (91.7, 94.0) | 1 | 1 |
| Ages 11–15 | | | | | |
| Trial participant | 1036 | 1038 | 99.8 (99.5, 1.00) | 0.6 (0.0 to 1.2) | 0.4 (−5.2 to 5.9) |
| Non-trial participant | 1115 | 1124 | 99.2 (98.6, 99.7) | 1 | 1 |
| Ages 16–17 | | | | | |
| Trial participant | 532 | 539 | 98.7 (97.8, 99.7) | 6.9 (4.6 to 9.2) | 6.3 (0.5 to 12.2) |
| Non-trial participant | 593 | 646 | 91.8 (89.7, 93.9) | 1 | 1 |
| Ages 18–20 | | | | | |
| Trial participant | 69 | 74 | 93.2 (87.5, 99.0) | 26.7 (18.5 to 34.9) | 19.5 (5.6 to 33.3) |
| Non-trial participant | 163 | 245 | 66.5 (60.6, 72.4) | 1 | 1 |
| **2015: full cohort** | | | | | |
| All ages | | | | | |
| Trial participant | 1234 | 1524 | 81.0 (79.0, 83.0) | 12.5 (9.6 to 15.4) | 6.8 (3.4 to 10.2) |
| Non-trial participant | 1231 | 1797 | 68.5 (66.4, 70.7) | 1 | 1 |
| Ages 11–15 | | | | | |
| Trial participant | 837 | 952 | 87.9 (85.9, 90.0) | 3.8 (0.1 to 6.9) | 1.9 (− 1.3 to 5.0) |
| Non-trial participant | 833 | 990 | 84.1 (81.9, 86.5) | 1 | 1 |
| Ages 16–17 | | | | | |
| Trial participant | 361 | 503 | 71.8 (67.9, 75.8) | 12.5 (6.9 to 18.1) | 12.7 (6.7 to 18.7) |
| Non-trial participant | 346 | 584 | 59.3 (55.4, 63.4) | 1 | 1 |
| Ages 18–20 | | | | | |
| Trial participant | 36 | 69 | 52.2 (41.6, 65.4) | 28.9 (15.8 to 41.9) | 29.7 (16.1 to 43.3) |
| Non-trial participant | 52 | 223 | 23.3 (18.4, 29.6) | 1 | 1 |
| **2015: restricted cohort†** | | | | | |
| All ages | | | | | |
| Trial participant | 1234 | 1510 | 81.7 (79.8, 83.7) | 7.3 (4.5 to 10.2) | 4.9 (1.5 to 8.3) |
| Non-trial participant | 1231 | 1655 | 74.4 (72.3, 76.5) | 1 | 1 |
| Ages 11–15 | | | | | |
| Trial participant | 837 | 950 | 88.1 (86.1, 90.2) | 3.2 (0.2 to 6.3) | 1.8 (−1.3 to 5.0) |
| Non-trial participant | 833 | 982 | 84.8 (82.6, 87.1) | 1 | 1 |
| Ages 16–17 | | | | | |
| Trial participant | 361 | 496 | 72.8 (69.0, 76.8) | 7.6 (1.9 to 13.3) | 7.9 (1.9 to 14.0) |
| Non-trial participant | 346 | 531 | 65.2 (61.2, 69.3) | 1 | 1 |
| Ages 18–20 | | | | | |
| Trial participant | 36 | 64 | 56.3 (45.3, 69.8) | 19.6 (5.1 to 34.1) | 22.8 (7.3 to 38.2) |
| Non-trial participant | 52 | 142 | 36.6 (29.5, 45.5) | 1 | 1 |

*Adjusted for socioeconomic status (coded dichotomously at median household asset index score), country of origin (South African vs Mozambican), educational attainment of household head (coded dichotomously at grade 12 attainment), gender of household head (male vs female), household size (coded linearly) and pre-2011 childbearing (yes vs no). Models that are not age-stratified are also adjusted for age coded in 2-year categories.
†Restricted to all young women who were enrolled in school in 2011.
aRD, adjusted risk difference; HPTN, HIV Prevention Trial Network; RD, risk difference.

**Table 3** Sensitivity analysis comparing differences in school enrolment between HPTN 068 trial participants in the control arm only and non-trial participants

| | Enrolment | n | Per cent enrolled | RD (95% CI) | aRD* (95% CI) |
|---|---|---|---|---|---|
| **2011: full cohort** | | | | | |
| All ages | | | | | |
| Trial participant (control) | 862 | 872 | 98.9 (98.2, 99.6) | 6.0 (4.7 to 7.3) | 3.2 (−1.4 to 7.7) |
| Non-trial participant | 1871 | 2015 | 92.9 (91.7, 94.0) | 1 | 1 |
| Ages 11–15 | | | | | |
| Trial participant (control) | 543 | 544 | 99.8 (99.5, 1.00) | 0.6 (0.0 to 1.3) | 0.4 (−7.3 to 8.0) |
| Non-trial participant | 1115 | 1124 | 99.2 (98.6, 99.7) | 1 | 1 |
| Ages 16–17 | | | | | |
| Trial participant (control) | 280 | 285 | 98.3 (96.7, 99.8) | 6.5 (3.8 to 9.1) | 5.7 (−0.9 to 12.3) |
| Non-trial participant | 593 | 646 | 91.8 (89.7, 93.9) | 1 | 1 |
| Ages 18–20 | | | | | |
| Trial participant (control) | 39 | 43 | 90.7 (82.0, 99.4) | 24.2 (13.7 to 34.7) | 17.7 (2.6 to 32.8) |
| Non-trial participant | 163 | 245 | 66.5 (60.6, 72.4) | 1 | 1 |
| **2015: full cohort** | | | | | |
| All ages | | | | | |
| Trial participant (control) | 653 | 813 | 80.3 (77.6, 83.1) | 11.8 (8.3 to 15.3) | 6.9 (3.6 to 10.3) |
| Non-trial participant | 1231 | 1797 | 68.5 (66.4, 70.7) | 1 | 1 |
| Ages 11–15 | | | | | |
| Trial participant (control) | 448 | 504 | 88.9 (86.2, 91.6) | 4.8 (1.2 to 8.3) | 2.4 (−1.3 to 6.1) |
| Non-trial participant | 833 | 990 | 84.1 (81.9, 86.5) | 1 | 1 |
| Ages 16–17 | | | | | |
| Trial participant (control) | 186 | 269 | 69.1 (63.6, 74.7) | 9.9 (3.1 to 16.7) | 9.7 (2.6 to 16.8) |
| Non-trial participant | 346 | 584 | 59.3 (55.4, 63.4) | 1 | 1 |
| Ages 18–20 | | | | | |
| Trial participant (control) | 19 | 40 | 47.5 (32.0, 63.0) | 24.2 (7.7 to 40.6) | 21.5 (4.9 to 38.1) |
| Non-trial participant | 52 | 223 | 23.3 (18.4, 29.6) | 1 | 1 |
| **2015: restricted cohort†** | | | | | |
| All ages | | | | | |
| Trial participant (control) | 653 | 803 | 81.3 (78.6, 84.0) | 6.9 (3.5 to 10.4) | 4.5 (1.3 to 7.6) |
| Non-trial participant | 1231 | 1655 | 74.4 (72.3, 76.5) | 1 | 1 |
| Ages 11–15 | | | | | |
| Trial participant (control) | 448 | 503 | 89.1 (86.3, 91.8) | 4.2 (0.7 to 7.8) | 2.5 (−1.2 to 6.2) |
| Non-trial participant | 833 | 982 | 84.8 (82.6, 87.1) | 1 | 1 |
| Ages 16–17 | | | | | |
| Trial participant (control) | 186 | 264 | 70.5 (65.0, 76.0) | 5.3 (−1.5 to 12.1) | 5.2 (−2.0 to 12.3) |
| Non-trial participant | 346 | 531 | 65.2 (61.2, 69.3) | 1 | 1 |
| Ages 18–20 | | | | | |
| Trial participant (control) | 19 | 36 | 52.8 (36.5, 69.1) | 16.2 (−2.0 to 34.3) | 13.8 (−5.7 to 33.4) |
| Non-trial participant | 52 | 142 | 36.6 (29.5, 45.5) | 1 | 1 |

School enrolment outcome was analysed in 2011 and 2015 and the analysis was stratified by age, in the full cohort and the cohort restricted to those enrolled in school in 2011.

*Adjusted for socioeconomic status (coded dichotomously at median household asset index score), country of origin (South African vs Mozambican), educational attainment of household head (coded dichotomously at grade 12 attainment), gender of household head (male vs female), household size (coded linearly) and pre-2011 childbearing (yes vs no). Models that are not age-stratified are also adjusted for age coded in 2-year categories.

†Restricted to all young women who were enrolled in school in 2011.

aRD, adjusted risk difference; HPTN, HIV Prevention Trial Network; RD, risk difference.

differences observed in the youngest age group are understandable as they were under the age limit of compulsory education with requirements to remain in school regardless of trial influence. For the oldest age group, trial selection criteria for lower grade levels meant they were older than expected for their grade, and suggested a history of grade repetition or temporary dropout. That the trial protocol may have contributed to keeping older teens in school is significant as the transition to adulthood carries extremely high HIV risk.[23]

This study was fairly unusual in that data were available on key study outcomes for the underlying population from which study participants were drawn. The Agincourt HDSS routinely collects school enrolment data on all residents in the study area, and we were able to leverage those data to assess differences between trial participants and non-participants. The majority of epidemiological studies do not have the benefit of complete background data on the target population, and, as such, sample selection and Hawthorne effects are rarely empirically assessed.[5 10 12 14] However, when Hawthorne effects are assessed, the direction of the relationship between research participation and healthy outcomes tends to be positive, in line with our findings of improved school enrolment outcomes.

It is important to note that data on HIV incidence, the primary endpoint of HPTN 068, were not available for the underlying target population. We speculate that the improved schooling trajectories we observed in trial participants likely resulted in reduced risk of HIV acquisition.[17] Continued schooling is strongly associated with HIV prevention and reduced sexual risk outcomes in young women in sub-Saharan Africa,[18 27 28] and we observed lower HIV incidence (1.8%) than expected among trial participants (3%). However, we cannot say with certainty that the association between trial participation and school enrolment extended to HIV protection.

The potential for uncontrolled confounding requires that our results be interpreted cautiously. The differences we attribute to the Hawthorne effect were estimated in an observational data set. Initial screens for trial eligibility were performed based on age data maintained by the Agincourt HDSS, and 82% of the eligible young women approached went on to enrol in the study, a fairly high response rate.[17] Still, it is plausible that those who refused participation were different from those who consented in ways that were also related to future school enrolment trajectories. Although we controlled for key sociodemographic characteristics that we theorised could be related to both trial participation and school enrolment, the possibility for bias from unmeasured confounding remains.

We offer three key conclusions from this study. First, epidemiologists should give greater weight at the planning, analysis and dissemination stages to identifying how sample selection and Hawthorne effects can be minimised, assessed and discussed. Prioritising research with well-defined target populations in areas with ongoing background data collection (eg, HDSS centres) would improve researchers' abilities to empirically assess the external validity of their findings. Second, the sample selection effect we observed highlights how school-based samples can differ in important ways from non-school-based samples in terms of underlying risk. Interventions focused on school-going adolescents may not reach those most in need of prevention, an anticipated issue that was ultimately difficult to avoid given the HPTN 068 design. Third, the Hawthorne-specific findings suggest that aspects of the HPTN 068 protocol could potentially be adapted for school retention interventions to prevent HIV. If the relationship we observed is causal, the trial protocol increased school enrolment at a magnitude similar to targeted cash transfer interventions and other fairly resource-intensive school retention interventions in sub-Saharan Africa,[21 29–31] despite the actual contact with the young women being limited to annual visits. Future work should examine key elements of the study protocol—adult mentorship, peer support, school attendance monitoring, messaging around the link between school and HIV, routine HIV/sexually transmitted infection testing and linkage to care—to better understand their relationship with school retention and HIV acquisition.

**Author affiliations**
[1]Department of Epidemiology and Biostatistics, Indiana University School of Public Health, Bloomington, Indiana, USA
[2]Department of Epidemiology, University of North Carolina, Chapel Hill, North Carolina, USA
[3]Carolina Population Center, University of North Carolina, Chapel Hill, North Carolina, USA
[4]MRC/Wits Rural Public Health and Health Transitions Research Unit (Agincourt), Faculty of Health Sciences, School of Public Health, University of the Witwatersrand, Johannesburg, South Africa
[5]Department of Biostatistics, University of Washington, Seattle, Washington, USA
[6]Center for Population and Development Studies, Harvard University, Cambridge, Massachusetts, USA
[7]INDEPTH Network, Accra, Ghana
[8]Division of Epidemiology and Global Health, Department of Public Health and Clinical Medicine, Umeå Centre for Global Health Research, Umeå University, Umeå, Sweden
[9]School of Health and Society, University of Wollongong, Wollongong, New South Wales, Australia
[10]Wits Reproductive Health and HIV Institute, University of the Witwatersrand, Johannesburg, South Africa

**Contributors** MR, AP, RT and KK conceived the study. MR conducted the analysis and wrote the first draft of the manuscript. AP, RT, JPH, FXG-O, RGW, ASu, ST, ASe, CM and KK were involved in the design of the parent studies and/or in collection, storage and analysis of data from the parent study. All authors contributed to the interpretation of the findings, critical review of the manuscript and approval of the final manuscript as submitted.

**Funding** The Agincourt Health and socio-Demographic Surveillance System is supported by the Wellcome Trust (058893/Z/99/A; 069683/Z/02/Z; 085477/Z/08/Z; 085477/B/08/Z), the University of the Witwatersrand and Medical Research Council, South Africa. HPTN 068 was supported by Award Numbers UM1 AI068619 (HPTN Leadership and Operations Center), UM1AI068617 (HPTN Statistical and Data Management Center) and UM1AI068613 (HPTN Laboratory Center) from the National Institute of Allergy and Infectious Diseases, the National Institute of Mental Health and the National Institute on Drug Abuse of the National Institutes of Health. This work was also supported by NIMH (R01MH087118) and the Carolina Population Center and its NIH Center grant (P2C HD050924).

**Disclaimer** The content is solely the responsibility of the authors and does not necessarily represent the official views of the National Institutes of Health.

**Competing interests**  None declared.

**Ethics approval**  Ethical approval was obtained from the University of the Witwatersrand's Human Research Ethics Committee (updated #M110138; original #M960720) and the Mpumalanga Province Health Research Committee. Ethical approvals for HPTN 068 were provided by the Office of Human Research Ethics at the University of North Carolina-Chapel Hill (#10–1868), the University of the Witwatersrand's Human Research Ethics Committee (#101012), and the Mpumalanga Province Health Research Committee. Ethical approval for this analysis was provided by the Indiana University Office of Research Compliance (#1608116129).

**Provenance and peer review**  Not commissioned; externally peer reviewed.

**Data sharing statement**  Agincourt HDSS data access can be requested at the following link: http://www.agincourt.co.za/index.php/data/.

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
