## [Reviewer comments · BMJ Open]

ARTICLE DETAILS

TITLE (PROVISIONAL)	Evidence for sample selection effect and Hawthorne effect in behavioral HIV prevention trial among young women in a rural South African community
AUTHORS	Rosenberg, Molly; Pettifor, Audrey; Twine, Rhian; Hughes, James; Gómez-Olivé, F. Xavier; Wagner, Ryan; Sulaimon, Afolabi; Tollman, Stephen; Selin, Amanda; MacPhail, Catherine; Kahn, Kathleen

VERSION 1 – REVIEW

REVIEWER	Miriam Fenton Evidera, United States
REVIEW RETURNED	13-Sep-2017

GENERAL COMMENTS	This is a very interesting and unusual study. The authors leverage a unique situation- knowledge of the source population, in order to assess the Hawthorne effect. Specific comments are below. Intro - First sentence. This is not a non-contentious statement, and warrants, at the very least, some references.- This is not an epi journal. Too much basic epi knowledge is assumed of the reader. Epi concepts such as “external validity” should be defined and explained. Method - “Other socio-demographic data is collected at regular but less frequent intervals.”o It would be helpful if this was quantified- The methods should include information on the trial procedure. “HPTN 068 was a Phase III individually-randomized trial designed to examine whether cash transfers conditional on school attendance influence the risk of HIV acquisition in young women” is not an adequate explanation. At the very least readers should be referred to articles with a more thorough description. Results - Confidence intervals for adjusted estimates are extremely large, which limits confidence in these results- This section is not entirely clear, largely because we do not know the trial procedures. Explaining this in the methods would help the reader better follow the results section. Conclusions - More information on the implications of these results on trial findings would be useful
--

REVIEWER	Salim S. Abdool Karim CAPRISA, South Africa
REVIEW RETURNED	18-Sep-2017

GENERAL COMMENTS	By constructing a cohort from census data in Agincourt, Rosenberg and colleagues were able to compare the unmarried (and not pregnant) 13-20 year-old women from a community-based census with those who had been enrolled in a cash-incentive study in this same community. The differences they found at the beginning of the trial between the census cohort and the trial enrolees with regard to school attendance, socio-economic status, child-bearing initiation, amongst others, indicated sampling bias. While differences, such as school attendance, between trial enrolees and the community cohort at the end of the trial were ascribed to a Hawthorne effect. Major comments:  1. The investigators are applauded for using the unique opportunity of conducting a trial in a community with repeat census data, to study differences between the population and the study participants. 2. While differences were not noted between trial enrolees and the community, these sampling biases as not directly relevant to the trial outcome as long as the women in the two trial arms were comparable ie. a trial does not need to reflect the underlying community in order to be valid. Indeed, trials generally aim to select their participants from a trial to increase the probably of showing the hypothesized difference. Since trial outcomes will need to be extrapolated to a range of different communities, trials generally do not aim to accurately represent a single community. Whereas the sampling biases do not impact internal validity of the trial and its results, the bias of lower school dropout among trial enrolees show how the trial's inclusion/exclusion criteria ended up inadvertently undermining its ability to produce an outcome by selecting women who were less likely to drop out of school. In other words, this finding points to an unforeseen design flaw in the trial's selection criteria rather than any real concern about external validity of the trial. 3. Ascribing the findings from the analysis comparing school dropout rates among the trial participants with non-participants as a Hawthorne effect is flawed and is not scientifically sound. Since the estimate of the Hawthorne effect is small and not statistically significant – it is not clear if this is driven by the intervention AND the Hawthorne effect or just the Hawthorne effect. Even though the 2 trial arms had similar HIV incidence rates, this is not sufficient grounds for merging the 2 groups in this analysis. Since the trial was designed to decrease school dropout, showing that school dropout at the end of the study was lower in trial participants is to be expected as a result of the intervention, and not necessarily a result of women feeling observed. The flaw in this analysis can be remedied by comparing only the control arm women from the trial with the non-participants from the community. Since the control arm women were observed, but did not have any specific interventions to reduce school dropout, comparing them to women who did not participate in the trial will provide a more reasonable approximation of the extent to which women reduced their school drop-out just as a result of trial participation and not trial intervention.
--

	4. The baseline comparison of school enrolment (2011) is made using school enrolment data projected from 2009. These findings hinge on these projections being accurate. In the key measures section, the authors provide a brief description of how this was done but given the importance of these projections, more detail is required here. For example, the authors say 'we used changes in educational attainment to infer whether school dropout had occurred before or after 2011', with no explanation of what they mean by 'changes in educational attainment', how it was measured and how accurate it is likely to be. 5. There are clear differences in the trial participant group and the non-trial participant group – by age, SES etc – so it would be more appropriate to look at the adjusted RD results. But in the abstract, the authors have reported the 2011 unadjusted RD values which are large and significant - this is misleading. The authors need to provide the results in the abstract that support their conclusion that Hawthorne effects were observed – there are no data in the abstract related to the Hawthorne effect. Overall, this is an interesting analysis. However, the findings do not support the author's main conclusions – the main outcome is the identification of an inclusion criterion flaw rather than any real concern about sampling bias and the secondary outcome on the Hawthorne effect can be assessed using the approach here; the control arm women need to be compared with women who did not participate in the trial to assess if there is any Hawthorne effect.
--	---

VERSION 1 – AUTHOR RESPONSE

Reviewer 1:

1. First sentence. This is not a non-contentious statement, and warrants, at the very least, some references.

Response: We agree with the reviewer that caution and justification is warranted with this sentence. We have now added two citations to support the idea that epidemiologic studies are used to build the evidence-base for public health interventions.

2. This is not an epi journal. Too much basic epi knowledge is assumed of the reader. Epi concepts such as "external validity" should be defined and explained.

Response: We now define external validity with its synonym 'generalizability' when it is first introduced in the paragraph.

We further attempt to provide a definition of both 'sample selection effect' and 'Hawthorne effect' when they are both first introduced.

3. "Other socio-demographic data is collected at regular but less frequent intervals." It would be helpful if this was quantified

Response: We now include three examples of the frequency with which key variables are collected in the Agincourt HDSS:

"For example, educational attainment is queried every three years, employment data are collected every four years, and a household asset index is measured every other year."

4. The methods should include information on the trial procedure. “HPTN 068 was a Phase III individually-randomized trial designed to examine whether cash transfers conditional on school attendance influence the risk of HIV acquisition in young women” is not an adequate explanation. At the very least readers should be referred to articles with a more thorough description.

Response: We now include more details about the cash transfer protocol in the second paragraph of the Methods section:

“Young women and their caregivers were randomly assigned to receive a monthly cash transfer conditional on $\geq 80\%$ school attendance or no cash transfer. The size of the monthly cash transfer was 300 Rands (R; about US\$30 in 2012), and was divided into R200 provided to the caregiver and R100 provided to the young woman.”

We also include reference to two publications which describe the HPTN 068 trial protocol in detail:

1. Pettifor A, MacPhail C, Selin A, et al. HPTN 068: a randomized control trial of a conditional cash transfer to reduce HIV infection in young women in South Africa—study design and baseline results. *AIDS and behavior* 2016;20(9):1863-82.

2. Pettifor A, MacPhail C, Hughes JP, et al. The effect of a conditional cash transfer on HIV incidence in young women in rural South Africa (HPTN 068): a phase 3, randomised controlled trial. *The Lancet Global Health* 2016;4(12):e978-e88.

5. Confidence intervals for adjusted estimates are extremely large, which limits confidence in these results

Response: Due to the constraints we were placing on the data (adjusting for a broad set of sociodemographic characteristics, stratifying fairly finely by age, etc), it is perhaps not surprising that some of our results are not very precise. We agree with the reviewer that this imprecision could lead to limited confidence in some of our results.

The widest confidence intervals are those around the effect estimates of the oldest age group (age 18-20 at baseline). This group had limited sample size because it was uncommon for young women this old to still be enrolled in the grades targeted by HPTN 068 (Grades 8-11). The limited sample size of this group is likely directly responsible for the wide confidence intervals. The overall adjusted estimate and the adjusted estimates for the younger ages are more precise. However, because the estimates for the older age group are so much larger in magnitude than those in the younger age groups, we thought it important to report the age-stratified estimates with their associated 95% CIs.

6. This section is not entirely clear, largely because we do not know the trial procedures. Explaining this in the methods would help the reader better follow the results section.

Response: We hope that the additional details provided in the methods section as outlined in our response to Comment 4, and the detailed information available in the baseline and endline HPTN 068 trial papers we cite in the methods section will provide the information necessary to better understand the context of the parent trial and interpret the results of the current study.

7. More information on the implications of these results on trial findings would be useful

Response: We are unable to definitively state whether the sample selection effect and Hawthorne effect (for which we found evidence in this study) produced trial results that were different from those we would have found in the absence of sample selection and Hawthorne effects.

In that spirit, in the discussion, we speculate that:

“Both sample selection and Hawthorne effects may have diminished the differences in school enrollment between study arms and is one plausible explanation for the overall null study effect.”

And have now added:

“The HPTN 068 trial was designed to activate the HIV prevention effects of education by incentivizing school attendance and retention in the intervention arm. With high levels of school attendance and retention across both arms of the trial, the ability to detect a trial effect was likely weakened. “

Later in the third paragraph of the discussion, we also speculate that:

“...the improved schooling trajectories we observed in trial participants likely resulted in reduced risk of HIV acquisition.

But cautiously add that :

“However, we cannot say with certainty that the association between trial participation and school enrollment extended to HIV protection.”

Reviewer 2:

1. While differences were not noted between trial enrollees and the community, these sampling biases as not directly relevant to the trial outcome as long as the women in the two trial arms were comparable ie. a trial does not need to reflect the underlying community in order to be valid. Indeed, trials generally aim to select their participants from a trial to increase the probably of showing the hypothesized difference. Since trial outcomes will need to be extrapolated to a range of different communities, trials generally do not aim to accurately represent a single community. Whereas the sampling biases do not impact internal validity of the trial and its results, the bias of lower school dropout among trial enrollees show how the trial’s inclusion/exclusion criteria ended up inadvertently undermining its ability to produce an outcome by selecting women who were less likely to drop out of school. In other words, this finding points to an unforeseen design flaw in the trial’s selection criteria rather than any real concern about external validity of the trial.

Response: We agree with the reviewer that there are likely not many threats to the internal validity of the HPTN 068 trial. The cash transfer intervention was randomized across a fairly large sample of young women, delivered with fidelity, and evaluated with rigorously collected laboratory-based outcomes.

But we do posit that the external validity of the trial is exactly what is threatened by the sample selection effect and Hawthorne effect we identify in this study. The young women who enrolled in the trial differed in their school enrollment behaviors compared to the general population of rural South African young women to whom we would like to generalize the trial findings. Our findings in this study suggest that the difference in school enrollment behaviors occurred both because better enrollers were selected into the trial (which perhaps could have been avoided with different inclusion/exclusion criteria) AND because being in the trial itself caused behavior change with respect to school enrollment (which is unlikely to have changed with different inclusion/exclusion criteria).

2. Ascribing the findings from the analysis comparing school dropout rates among the trial participants with non-participants as a Hawthorne effect is flawed and is not scientifically sound. Since the estimate of the Hawthorne effect is small and not statistically significant – it is not clear if this is driven by the intervention AND the Hawthorne effect or just the Hawthorne effect. Even though the 2 trial arms had similar HIV incidence rates, this is not sufficient grounds for merging the 2 groups in this analysis. Since the trial was designed to decrease school dropout, showing that school dropout at the end of the study was lower in trial participants is to be expected as a result of the intervention, and not necessarily a result of women feeling observed. The flaw in this analysis can be remedied by comparing only the control arm women from the trial with the non-participants from the community.

Since the control arm women were observed, but did not have any specific interventions to reduce school dropout, comparing them to women who did not participate in the trial will provide a more reasonable approximation of the extent to which women reduced their school drop-out just as a result of trial participation and not trial intervention.

Response: With apologies if this is unclear in the results section: the effect estimate we are attributing to the Hawthorne effect is statistically significant in both adjusted and unadjusted models. In our restricted cohort, the young women enrolled in the trial were more likely to remain in school in 2015 (82%), compared to those who did not (74%) with an adjusted risk difference of 4.9 (95% CI: 1.5, 8.3)]. All three of the age-stratified estimates were also statistically significant in unadjusted models and only the confidence intervals for the estimate for the youngest age category spanned the null after adjustment.

That said, we understand the concern around combining the control and intervention arms for analysis given the aim of the trial was to increase school attendance. Our justification for analyzing the two arms together was that trial results indicated essentially no differences in school attendance and enrollment data between the arms. 95% of the intervention arm and 96% of the control arm attended school >80% of school days. Permanent school dropout occurred at a rate of 3 per 100 person years in both arms.¹ We now provide this extra information in the first 'Key measures' paragraph in the Methods section.

As suggested, we also conducted a sensitivity analysis comparing the control arm participants to non-participants to see whether results differed when compared to those from the analysis with both trial arms combined. Trends in the results held for all of our key outcomes and subgroups (enrollment in 2011 and 2015, stratified by age) – see Table 3. Although confidence intervals widened due to reduced sample size (and some now span the null where they did not in the analysis with trial arms combined), the magnitudes of the risk difference point estimates were largely unchanged. We give details about this sensitivity analysis at the end of both the methods and results sections.

1 Pettifor A, MacPhail C, Hughes JP, et al. The effect of a conditional cash transfer on HIV incidence in young women in rural South Africa (HPTN 068): a phase 3, randomised controlled trial. *The Lancet Global Health* 2016;4(12):e978-e88.

3. The baseline comparison of school enrolment (2011) is made using school enrolment data projected from 2009. These findings hinge on these projections being accurate. In the key measures section, the authors provide a brief description of how this was done but given the importance of these projections, more detail is required here. For example, the authors say 'we used changes in educational attainment to infer whether school dropout had occurred before or after 2011', with no explanation of what they mean by 'changes in educational attainment', how it was measured and how accurate it is likely to be.

Response: We thank the reviewer for pointing out the need for more details around the 2011 school enrollment measure definition.

We expanded the third paragraph of the Key Measures section of the Methods section to include more details and intuition on how we coded the 2011 enrollment data based on information collected in 2009 and 2012. In addition to more information on the data collected in each education module, we added two practical examples of how we coded 2011 enrollment depending on data configurations as follows:

“For example, if a young woman reported grade 7 attainment in 2009 and grade 10 attainment in 2012, we assumed that the three years of additional education were accumulated in 2009, 2010, and 2011. This young woman would be coded as enrolled in school in 2011. If a young woman reported grade 7 attainment in 2009 and grade 9 attainment in 2012, we assumed that the two years of additional education were accumulated in 2009 and 2010. This young woman would be coded as not enrolled in school in 2011.”

4. There are clear differences in the trial participant group and the non-trial participant group – by age, SES etc – so it would be more appropriate to look at the adjusted RD results. But in the abstract, the authors have reported the 2011 unadjusted RD values which are large and significant - this is misleading. The authors need to provide the results in the abstract that support their conclusion that Hawthorne effects were observed – there are no data in the abstract related to the Hawthorne effect.
Response: We agree with the reviewer that the adjusted estimates for the Hawthorne-specific results would be more appropriate to present in the abstract and have edited it to include them.

5. Overall, this is an interesting analysis. However, the findings do not support the author’s main conclusions – the main outcome is the identification of an inclusion criterion flaw rather than any real concern about sampling bias and the secondary outcome on the Hawthorne effect can be assessed using the approach here; the control arm women need to be compared with women who did not participate in the trial to assess if there is any Hawthorne effect.
Response: We hope that the additional sensitivity analysis restricting to the comparison between control arm and non-trial participants and the responses we provided to the other reviewer comments alleviates your concerns around the robustness of our findings.

VERSION 2 – REVIEW

REVIEWER	Miriam Fenton Evidera, USA
REVIEW RETURNED	15-Nov-2017
GENERAL COMMENTS	The authors have addressed all of my comments appropriately. This is a very good manuscript.
REVIEWER	Salim S. Abdool Karim CAPRISA, South Africa and Columbia University, USA
REVIEW RETURNED	09-Nov-2017
GENERAL COMMENTS	The authors have addressed my points to the extent it is possible so I am prepared to approve this manuscript.